# Astaxanthin Prevents Atrophy in Slow Muscle Fibers by Inhibiting Mitochondrial Reactive Oxygen Species via a Mitochondria-Mediated Apoptosis Pathway

**DOI:** 10.3390/nu13020379

**Published:** 2021-01-26

**Authors:** Luchuanyang Sun, Nobuyuki Miyaji, Min Yang, Edward M. Mills, Shigeto Taniyama, Takayuki Uchida, Takeshi Nikawa, Jifeng Li, Jie Shi, Katsuyasu Tachibana, Katsuya Hirasaka

**Affiliations:** 1Graduate School of Fisheries and Environmental Sciences, Nagasaki University, Nagasaki 8528521, Japan; slcywelcome@126.com (L.S.); 18342221707@163.com (M.Y.); tshigeto@nagasaki-u.ac.jp (S.T.); orenge@nagasaki-u.ac.jp (K.T.); 2Toyo Koso Kagaku Co., Ltd., Chiba 2790041, Japan; miyaji@toyokk.co.jp; 3Division of Pharmacology/Toxicology, University of Texas at Austin, Austin, TX 78712, USA; tedmills@austin.utexas.edu; 4Department of Nutritional Physiology, Institute of Medical Nutrition, Tokushima University Medical School, Tokushima 7708503, Japan; t.uchida@tokushima-u.ac.jp (T.U.); nikawa@tokushima-u.ac.jp (T.N.); 5Weihai Lida Biological Technology Co., Ltd., Weihai 264200, China; lijifeng6812@gmail.com (J.L.); shijie.lida@gmail.com (J.S.); 6Organization for Marine Science and Technology, Nagasaki University, Nagasaki 8528521, Japan

**Keywords:** astaxanthin, muscle atrophy, mitochondria, oxidative stress

## Abstract

Astaxanthin (AX) is a carotenoid that exerts potent antioxidant activity and acts in the lipid bilayer. This study aimed to investigate the effects of AX on muscle-atrophy-mediated disturbance of mitochondria, which have a lipid bilayer. Tail suspension was used to establish a muscle-atrophied mouse model. AX diet fed to tail-suspension mice prevented loss of muscle weight, inhibited the decrease of myofiber size, and restrained the increase of hydrogen peroxide (H_2_O_2_) production in the soleus muscle. Additionally, AX improved downregulation of mitochondrial respiratory chain complexes I and III in the soleus muscle after tail suspension. Meanwhile, AX promoted mitochondrial biogenesis by upregulating the expressions of *adenosine 5′-monophosphate–activated protein kinase (AMPK) α-1*, *peroxisome proliferator–activated receptor (PPAR)-γ*, *and creatine kinase in mitochondrial (Ckmt) 2* in the soleus muscle of tail-suspension mice. To confirm the AX phenotype in the soleus muscle, we examined its effects on mitochondria using Sol8 myotubes derived from the soleus muscle. We found that AX was preferentially detected in the mitochondrial fraction; it significantly suppressed mitochondrial reactive oxygen species (ROS) production in Sol8 myotubes. Moreover, AX inhibited the activation of caspase 3 via inhibiting the release of cytochrome c into the cytosol in antimycin A–treated Sol8 myotubes. These results suggested that AX protected the functional stability of mitochondria, alleviated mitochondrial oxidative stress and mitochondria-mediated apoptosis, and thus, prevented muscle atrophy.

## 1. Introduction

Skeletal muscle atrophy has been observed in muscle disuse during unloading, immobilization, denervation, fasting, aging, and several disease conditions. Unloading-related muscle loss caused by prolonged bed rest or spaceflight specifically occurs in antigravity muscles including in slow muscle fibers [1,2]. It has been known that the number of mitochondria in slow muscle fibers is higher than that in fast muscle fibers [3]. Mitochondria are the main energy source of skeletal muscles that produce adenosine triphosphate (ATP) through oxidative phosphorylation (OXPHOS). In this process, 0.2–2.0% of diatomic oxygen passes through the electron transport chain complexes I and III and incompletely reduces leaked electrons to superoxide anions [4]. Mitochondria play a vital role in disused skeletal muscle atrophy [5] and mitochondrial signaling can contribute to disuse muscle atrophy in three major ways [6,7,8]: (1) increased mitochondrial reactive oxygen species (ROS) production, (2) energy stress: decrease in ATP production and the activation of adenosine 5′-monophosphate–activated protein kinase (AMPK), (3) mitochondria release of pro-apoptotic factors: release of cytochrome c into the cytosol and active caspase 3. In fact, the increase of mitochondrial ROS production could participate in both the energy stress and apoptosis [9]. Therefore, mitochondrial ROS has major impact on disuse-related muscle atrophy, and the development of antioxidants to prevent the increase in mitochondrial ROS in muscle due to inactivity has been extensively researched.

Astaxanthin (AX; 3,3′-dihydroxy-β, β′-carotene-4,4′-dione), a ketone carotenoid with 13 conjugated double bonds, naturally accumulates in microalgae, yeast, crustaceans, fish epidermis, and other biologicals. Natural AX is mainly obtained from *Haematococcus pluvialis*, and its biological effects as an antioxidant have been widely studied [10]. The unique molecular structure of AX allows for its insertion through the lipid bimolecular layer of cell membranes, leading to stronger protection and free-radical-scavenging effects at the cell membrane than other antioxidants such as β-carotene, α-tocopherol, and vitamin C [11,12]. AX reportedly maintains mitochondrial integrity by reducing oxidative stress, prevents the loss of mitochondrial membrane potential, and increases mitochondrial oxygen consumption, which inhibits mitochondrial dysfunction [13,14,15]. Additionally, AX suppresses bleomycin-induced ROS generation and apoptosis mediated by the disturbed mitochondrial signaling pathway in type II alveolar epithelial cells [16]. These reports raise the possibility that AX acts in both the mitochondrial and cell membranes.

Although antioxidants have been shown to prevent muscle atrophy [17,18], their mechanisms of action vary owing to their different characteristics. Some studies have shown that AX might have an effect on muscle atrophy [19,20]; however, its mechanism of action in mitochondria remains unclear. This study investigated the effect of AX on muscle atrophy induced by mitochondrial oxidative stress and dysfunction. Sol8 cells (slow-type muscle cells extracted from the soleus muscle) were used to explore the underlying mechanisms.

## 2. Materials and Methods

### 2.1. Animal Model

Male C57BL/6J mice (Japan CLEA, Tokyo, Japan) (6 weeks old) were housed in a room maintained at 24 ± 1 °C and a 12 h light/dark cycle with food (Oriental Yeast Company, Tokyo, Japan) and water available ad libitum. After 1 week of acclimatation, mice were randomized into four groups: control mice fed the normal diet (C-ND, *n* = 6); tail-suspension mice fed the normal diet (S-ND, *n* = 6); control mice fed the AX diet (C-AX, *n* = 6); and tail-suspension mice fed the AX diet (S-AX, *n* = 6). An AX-supplemented diet (0.2%, *w*/*w*) or a normal diet was fed to mice for four weeks. Since then, these mice (11 weeks old) were subjected to tail suspension to establish the muscle atrophy model. Briefly, the mouse tail was fixed with medical tape; the other end of the tape was attached to the top of the cage to keep its body at a 30° angle with the surface. The forelimbs were free to move on the ground to enable free access to water. During the development of tail suspension–induced muscle atrophy, the mice continued to receive normal or AX-supplemented diets until termination of the experiment two weeks later. The soybean oil content of the AX-supplemented diet was reduced to adjust for the composition of other nutrients. The AX-supplemented diet comprised the normal diet (based on AIN-93G) mixed with BioAstin SCE (containing 10% AX; Toyo Koso Kagaku Co. Ltd., Chiba, Japan). The nutritional composition of each diet is shown in Appendix A, the right hindlimb skeletal muscles including the tibialis anterior (TA), extensor digitorum longus (EDL), gastrocnemius (GA), and soleus (SO) muscles were isolated at the time of sacrifice. After measuring the wet weight, the skeletal muscles were immediately frozen in chilled isopentane with liquid nitrogen and stored at −80 °C until analysis. All animal experiments were approved by the Committee on Animal Experiments of Nagasaki University (permission no. 1803291443) and performed according to the guidelines for the care and use of laboratory animals set by the University.

### 2.2. Multicolor Immunofluorescence Staining and Measurement of Cross-Sectional Area (CSA)

The separated SO muscle was placed on the gum tragacanth and immediately frozen in chilled isopentane with liquid nitrogen and stored at −80 °C until analysis. The mid-belly transverse cryosections of SO muscles (5 μm thickness) were made with LEICA CM1950 cryostat (Wetzlar, Germany). Sections were put on poly-L-lysine-coated slide glasses, fixed in ice-cold acetone, and stained using multicolor immunofluorescence antibodies as described previously [21]. Primary antibody reactions were performed using anti-myosin heavy chain (MHC) type I (BA-F8), anti-MHC IIa (SC-71), and anti-MHC IIb (BF-F3) (DSHB, Iowa City, IA, USA); the secondary antibodies used were anti-mouse Alexa Fluor 350 IgG_2b_, anti-mouse Alexa Fluor 488 IgG_1_, and anti-mouse Alexa Fluor 555 IgM, respectively (Thermo Fisher Scientific, Waltham, MA, USA). Images were acquired using a BIOREVO BZ-X710 microscope (Keyence, Osaka, Japan). For fiber type analysis, all fibers within the entire cross-section were characterized. At least 600 myofiber cross-sectional areas (CSAs) were measured per group.

### 2.3. Cell Culture

Sol8 myoblastic cells were obtained from the American Type Culture Collection (Rockville, MD, USA). They were seeded in a collagen-coated plate and cultured in Dulbecco’s modified Eagle medium (DMEM) supplemented with 10% fetal bovine serum (FBS), 100 units/mL penicillin, and 100 μg/mL streptomycin and maintained at 37 °C in a 5% CO_2_ environment. At a confluence of 100%, Sol8 myoblasts were fused by shifting the medium to DMEM supplemented with 2% horse serum (HS). Cells were maintained in 2% HS/DMEM (differentiation medium) for 4 days for the formation of myotubes, as described previously [22].

### 2.4. Isolation of Mitochondria

Mitochondria from Sol8 cells, and skeletal muscle from C57BL6/j mice were prepared as previously [23,24]. Briefly, tissues or cells were minced in ice-cold CP-1 buffer (100 mM KCl, 50 mM Tris-HCl, 2 mM EGTA, pH 7.4). After homogenization by a plastic electric rod, then centrifuged at 500× *g* for 10 min. The supernatant was collected and further centrifuged for 10 min at 10,500× *g* to obtain the mitochondrial pellet and cytosolic fraction.

### 2.5. Detection of AX in Mitochondrial and Cytosolic Fractions

The AX contents in mitochondrial and cytosolic fractions from Sol8 myotubes were prepared, as described previously [23]. Briefly, Sol8 myotubes were treated with AX (100 nmol) or DMSO, as control, in 10 mL culture medium/100 mm dishes and incubated at 37 °C with 5% CO_2_ for 24 h. The cells were harvested to isolate the mitochondrial fraction as described in Section 2.4. After lyophilization, crude mitochondrial extracts and cytosol were solubilized in acetone and centrifuged at 12,000× *g* for 15 min. The supernatants were filtered using a 0.45 μm polytetrafluorethylene membrane and analyzed using HPLC and a spectrophotometer detector (JASCO, Tokyo, Japan) set at 460 nm. A Shim-pack CLC-ODS column (150 mm length and 6.0 mm internal diameter) was used.

### 2.6. Detection of H_2_O_2_ Production

The rate of H_2_O_2_ production by isolated mitochondria was detected using a fluorescent probe, Amplex Red, as described previously [24]. Mitochondria from muscle tissues were suspended in a buffer containing 5 mM 3-(N-morpholino)propanesulfonic acid (MOPS) (pH 7.4), 70 mM sucrose, and 220 mM mannitol, and mitochondria protein concentration was determined by using a bicinchoninic acid (BCA) protein assay (Pierce, Rockford, IL, USA). Mitochondria (30 μg mitochondrial protein per well) were incubated in black 96-well-plates with a reaction mixture containing 50 μM Amplex Red, 2 U/mL horseradish peroxidase, 30 U/mL superoxide dismutase (SOD), 10 mM succinate, 10 μM antimycin A as substrate at room temperature for 30 min, protected light. SOD was added to convert all superoxide into H_2_O_2_. SOD was added to convert all superoxide into H_2_O_2._ The rate of H_2_O_2_ production was linear with respect to mg of mitochondrial protein.

### 2.7. Measurement of Mitochondrial Superoxide Levels and Mitochondrial Membrane Potential (MMP)

Dihydroethidium (DHE) was used as a second method for the detection of superoxide. Differentiated Sol8 myotubes were plated in 96-well black plates (3 × 10^3^ cells/well) and treated with DMSO or AX for 24 h. After washing with HBSS, the cells were then incubated with HBSS containing 10 mM succinate, 10 μM antimycin A, and 5 μM DHE for 30 min. Fluorescence was recorded using a microplate reader (BioTek Cytation 3, Winooski, VT, USA) at excitation and emission wavelengths of 490 and 595 nm, respectively [24].

MMP was detected using the fluorescent probe, JC-1 dye (Thermo Fisher Scientific). Differentiated Sol8 myotubes were plated in 96-well black plates and pretreated with DMSO or AX for 1 h, followed by the addition of 25 μM antimycin A for 48 h. After washing with PBS, the cells were incubated with 1.5 μM JC-1 dye at 37 °C for 30 min. MMP was quantified using a microplate reader (BioTek Cytation 3) at 550 nm excitation/600 nm emission and 485 nm excitation/535 nm emission wavelengths. The quantitative MMP value was expressed as the relative ratio of aggregate-to-monomer values of fluorescence intensity. Carbonyl cyanide m-chlorophenyl hydrazone (CCCP) was used as a negative control to normalize changes in the membrane.

### 2.8. Real-Time Reverse Transcription (RT)–Polymerase Chain Reaction (PCR)

Total RNA was extracted from muscle using an acid guanidinium thiocyanate–phenol–chloroform mixture (ISOGEN; Nippon Gene, Tokyo, Japan). Real-time RT-PCR was performed with the appropriate primers and SYBR Green dye using a real-time PCR system (ABI Real-Time PCR Detection System; Applied Biosystems, Foster City, CA, USA), as described previously [25]. The oligonucleotide primers used for PCR are shown in Appendix A.

### 2.9. Immunoblotting

Mouse SO muscles and Sol8 were prepared in 50 mM Tris-HCl, pH 8.0, containing 1% NP40, 0.5% sodium deoxycholate, 0.1% SDS, 150 mM NaCl, 2 mM EDTA, and protease inhibitors (Roche Diagnostics, Tokyo, Japan) and homogenized using a sonicator. The cytosolic fraction for the analysis of cytochrome c release was prepared as described in Section 2.4. The Pierce BCA assay (Pierce, Rochford, IL, USA) was used to quantify proteins. Protein samples were combined with 4X sample buffer (250 mM Tris-HCl, 8% SDS, 40% glycerol, 8% beta-mercaptoethanol, and 0.02% bromophenol blue) and subjected to SDS-PAGE. The proteins were transferred onto a polyvinylidene difluoride (PVDF) membrane and probed with primary antibodies according to the manufacturer’s instructions. Anti-β-actin (Gene Tex, AC-15, Irvine, CA, USA), anti-cleaved caspase 3 (Cell Signaling Technology, #9661, Danvers, MA, USA), anti-caspase 3 (Cell Signaling Technology, #9662, Danvers, MA, USA), anti-cytochrome c (Santa Cruz Biotechnology, sc-13156, Santa Cruz, CA, USA), anti-glyceraldehyde 3-phosphate dehydrogenase (GAPDH) (Santa Cruz Biotechnology, sc-25778), anti-total OXPHOS (Abcam, ab110413, Cambridge, UK), and anti-VDAC1/Porin (Abcam, ab14734, Cambridge, UK) were used. Donkey anti-rabbit IgG at 1:5000 (GE Healthcare, Little Chalfont, UK) and sheep anti-mouse IgG at 1:5000 (GE Healthcare, Little Chalfont, UK) were used as the secondary antibodies. Membranes were developed using ImmunoStar^®^ Zeta Western blotting detection reagents (Fujifilm Wako Pure Chemical Corporation, Osaka, Japan). Immunocomplexes on the membrane were analyzed by Image J software (National Institutes of Health, Bethesda, MD, USA).

### 2.10. Statistical Analyses

All data were analyzed by one-way/two-way analysis of variance (ANOVA), using SPSS statistics software, followed by the Tukey test for individual differences between groups. *p*-values < 0.05 were considered to indicate significant differences.

## 3. Results

### 3.1. Effect of Dietary AX on Muscle Mass and Fiber Size in Tail-Suspension Mice

To examine the effect of AX on weight gain in mice, we compared the body weights and wet weights of several muscles in the normal control and tail-suspension mice fed a normal or AX-supplemented diet. From food intake data, we found that 0.55 ± 0.15 mg/day of AX was fed to each mouse. There was no significant difference between the normal and AX-supplemented diet groups in food intake (Figure 1a). Although the body weights of mice in the S-ND and S-AX groups significantly decreased after tail-suspension, these differences were not significant (Figure 1b).

Tail suspension significantly decreased the weights of TA, GA, and SO muscles, but not that of the EDL (Figure 2). In comparison to the S-ND group, the S-AX group showed inhibition of muscle-weight reduction only in the SO. In contrast, AX supplementation failed to prevent tail suspension–induced muscle atrophy in the EDL, TA, and GA muscles.

Next, we analyzed the cross-sectional area (CSA) of the SO muscle fiber. It has been reported that tail-suspension affects slow fiber type muscles, especially type I and IIa, but not fast fiber type ones [26]. Because the soleus muscle is made up of type I fiber (30.6% ± 2.2%), type IIa fiber (49.1% ± 1.2%), type IIx (11.8% ± 1.7%), and other types of fibers, we performed different fiber typing to investigate which type of fiber was influenced by AX [21]. Multicolor immunofluorescent staining showed that it comprised MHC type I (blue) and IIa (green) myofibers. Type IIb (red) myofibers were hardly detected in the SO muscle fibers. Myofibers in the S-ND group showed decreased CSA staining and comprised type I and IIa fibers (Figure 3a), compared to the C-ND group. In contrast, the CSAs of muscle fibers stained with type I and IIa in the S-AX mice were similar to those observed in the C-AX mice (Figure 3a). We confirmed the average CSA of types I and IIa fibers in the SO muscle. Consistent with the results that indicated a decrease in muscle weights in the S-ND group, the average CSA of type I and IIa muscle fibers in the S-ND group was significantly decreased, compared to that in the C-ND group. Meanwhile, AX prevented the decrease of CSA caused by tail suspension and resulted in the increase of CSA rather than the reduction of CSA (Figure 3b). In particular, tail suspension in the S-AX group showed inhibition of the CSA reduction of muscle fibers of MHC type I and IIa.

### 3.2. Effect of Dietary AX on H_2_O_2_ Production in the Muscle of Tail-Suspension Mice

Excessive ROS including superoxide anions produced by mitochondria play a vital role in disused skeletal muscle atrophy [5,6]. In the measuring the amount of H_2_O_2_ production by using Amplex Red, all superoxide ions have been converted into H_2_O_2_ by SOD. The amount of H_2_O_2_ production was significantly increased by tail suspension in normal diet mice (S-ND group). In contrast, the AX-supplemented diet inhibited the increase of H_2_O_2_ production in the S-AX group (Figure 4).

### 3.3. Effect of Dietary AX on Oxidative Phosphorylation Respiration in the Muscle of Tail-Suspension Mice

Denervation-induced muscle atrophy reportedly impairs oxidative phosphorylation complex proteins in mitochondria [27]. In addition, we found AX prevented the increase of H_2_O_2_ production caused by tail suspension. Thus, we investigated the effect of AX on the levels of oxidative phosphorylation–related protein complexes in the muscle. The amounts of complexes I and III in the SO muscle of the S-ND group were significantly lower than that of the C-ND group. In contrast, there were no significant changes between the protein complexes in the C-AX and S-AX groups (Figure 5).

### 3.4. Effect of AX on Mitochondrial Biogenesis in the Muscle of Tail-Suspension Mice

When muscles undergo immobilization and denervation, a series of undesirable changes occur in the mitochondria of the muscle, including inhibition of mitochondrial biogenesis [28]. Moreover, it has been reported that AX treatment stimulates mitochondrial biogenesis in the skeletal muscle [29]. Consistent with previous reports, we found that AMPKα-1 and peroxisome proliferator–activated receptor (PPAR)-γ mRNA expressions in the S-ND group were significantly decreased, compared with those in the C-ND group, whereas the S-AX group showed inhibition of the decrease of these mRNA expressions in SO muscle. In addition, the expression of Ckmt2 mRNA in the S-AX group was significantly higher than that in the S-ND group. Although there were no significant differences between S-ND and S-AX in the expression of *uncoupling protein 2 (Ucp2)*, *adenosine triphosphate 5g1 (Atp5g1)*, *NADH: ubiquinone oxidoreductase complex assembly factor (Ndufa) and succinate dehydrogenase complex*, *subunit B (Sdhb)*, AX slightly improved the expression in tail-suspension mice (Figure 6).

### 3.5. Effect of AX on Mitochondrial Function in Sol8 Myotubes

Manabe et al. reported that AX is more likely to accumulate in the mitochondria of mesangial cells [30]. This finding and the results of our in vivo experiments implied that AX may target the mitochondria. To investigate the effect of AX on mitochondrial function, we determined the location of AX in the mitochondria by using Sol8 myotubes derived from SO. Although differentiated Sol8 myotubes contained myoblasts, myotubes’ differentiation marker (slow type myosin heavy chain) was detected in Sol8 myotubes (data not shown). Therefore, these cells were used in subsequent experiments. We found that AX was preferentially detected in the mitochondrial fraction, which was approximately 1% that of the AX treatment concentration in Sol8 myotubes; DMSO (0 nmol AX) was detected neither in the mitochondria nor in the cytosol (Table 1). This result was consistent with a previous report that AX is more likely to accumulate in the mitochondria in mesangial cells [30].

Next, we analyzed the effect of AX on mitochondrial ROS generation and MMP levels in Sol8 myotubes treated with antimycin A (AnA), a complex III inhibitor. Mitochondrial ROS production in AnA-treated Sol8 myotubes was significantly increased compared to that in non-treated cells. The increased production of complex III–driven ROS was significantly suppressed by AX treatment in a dose-dependent manner (Figure 7a). Addition of AX in AnA non-treated Sol8 myotubes did not influence MMP (Figure 7b). AnA treatment significantly decreased the MMP in Sol8 myotubes relative to non-treated cells, resulting in the same MMP level as that of the negative control, with the uncoupler CCCP. Nevertheless, improvements in MMP levels were noticed in the AX-treated cells, although there were no significant differences between the different AX concentrations. These results suggested that AX pretreatment may inhibit mitochondrial ROS production and maintain the MMP in Sol8 myotubes. These results indicated that AX has a role in mitochondrial protection.

### 3.6. The Effect of AX on the Expression of Apoptosis-Related Proteins in AnA-Treated Sol8 Myotubes

To explore the mitochondria-mediated apoptotic mechanism of AX in AnA-treated Sol8 myotubes, the release of cytochrome c into the cytosol (Figure 8a) and activation of caspase 3 (Figure 8b) were examined. The amount of cytochrome c in AnA-treated Sol8 myotubes, without AX, in the cytosolic fraction was significantly increased compared to that of the DMSO-treated control group (Figure 8a). In contrast, AX inhibited the release of cytochrome c into the cytosol of AnA-treated Sol8 myotubes. Furthermore, AnA treatment tended to increase the amount of cleaved caspase 3 in Sol8 myotubes, whereas AX effectively decreased the amount of cleaved caspase 3 in AnA-treated Sol8 myotubes (Figure 8b).

## 4. Discussion

The novel findings of this research revealed that dietary AX supplementation attenuated the decrease in muscle mass and myofibers in the SO muscle, preventing mitochondrial dysfunction caused by oxidative stress. AX particularly inhibited the reduction of mitochondrial complexes I and III protein content and regulated mitochondrial oxidative phosphorylation and biogenesis in the SO muscle of tail-suspension mice. In addition, AX treatment mitigated the generation of mitochondrial ROS, cytochrome c release into the cytosol, and caspase 3 activation in Sol8 myotubes.

Although the body weight of mice decreased after tail suspension, EDL muscle weight did not influence this reduction. This meant that muscle atrophy induced by tail suspension was not due to decreased body weight stemming from starvation. Indeed, numerous studies demonstrate that the weight loss in EDL is hardly detectable in suspended animals [31,32]. Consistent with these findings, tail suspension induced a loss of skeletal muscle, including the GA, TA, and SO, but not the EDL muscle (Figure 2). Although GA and TA are mostly composed of fast-twitch fibers, they also contain intermediate muscle fibers such as IIa (GA: 20.9% ± 1.6%, TA: 18.2% ± 2.4%). In contrast, EDL is occupied by fast-twitch fibers such as IIb. Indeed, we found that the size of IIa in muscle fibers in the S-ND group showed significantly decreased, compared with the C-ND group in the SO muscle (Figure 3). Thus, muscle atrophy caused by tail suspension preferentially affected slow-type rather than fast-type muscles. Moreover, AX prevented atrophy in the muscle containing type I and IIa myofibers. These findings suggest that AX acts in slow-twitch and intermediate muscle fibers.

The CSA of type I and IIa myofibers in S-AX group significantly increased, compared with S-CN or other groups (Figure 3). SO muscle contains type IIx (11.8% ± 1.7%) and type IIb (3.1% ± 1.1%) as well as type I (30.6% ± 1.2%), type IIa (49.1% ± 1.2%) [21]. Given the influence of AX in type I and IIa muscle fibers, AX is likely to contribute to the transformation of type IIx to type I and IIa in SO muscle of tail-suspension mice, indicating increased CSA in S-AX. Likewise, TA muscle contains type IIx (44.7% ± 11.9%, CSA: 2186.0 ± 35.2 μm^2^), type I (0.6% ± 1.6%, CSA: 1501.4 ± 7.1 μm^2^) and type IIa (18.2% ± 2.4%, CSA: 1369.6 ± 22.4 μm^2^). Additionally, GA muscle contains type IIx (19.6% ± 2.1%, CSA: 2186.0 ± 35.2 μm^2^), type I (7.9% ± 0.5%, CSA: 1743.4 ± 28.2 μm^2^) and type IIa (41.6% ± 1.3%, CSA: 1346.2 ± 22.8 μm^2^) [21]. We found that the wet weight of TA and GA in the C-AX group was significantly decreased, compared with C-ND group. AX is presumed to cause the transformation of type IIx to type I and IIa muscle fibers, resulting in the shift to small-sized fibers. However, further studies are necessary to define this mechanism.

We found that the ratio of body weight/food intake decreased significantly at 5 and 6 weeks in the tail-suspension group (S-ND and S-AX). Previous studies showed that there was an increased reliance on carbohydrate metabolism for energy associated with muscle unloading [33]. In addition, shift of fiber phenotype was related to the downregulation of mitochondrial proteins and upregulation of glycolytic protein, suggesting a shift from oxidative to glycolytic metabolism [26]. Consistent with these results, we found the decreased ratio of body weight/food intake and the shift of type fibers caused by tail suspension. Additionally, we found that the mRNA expression of oxidative metabolism–related transcription factors was decreased in the tail-suspension group. Thus, these results showed that the decreased ratio of body weight/food intake was associated with the energy shift from oxidative to glycolytic metabolism.

Numerous studies have demonstrated that mitochondria play an important role in muscle atrophy [5,7,8,34,35]. In oxidative phosphorylation, mitochondrial respiratory chain complexes I and III are believed to be the major sites of ROS leakage, although other components of oxidative phosphorylation also contribute to the production of ROS in the mitochondria [36,37]. In this study, we found that the amount of H_2_O_2_ production was significantly increased by tail suspension in normal diet mice (S-ND group). However, the AX-supplemented diet inhibited the increase of H_2_O_2_ production in the S-AX group. In addition, there was a significant reduction in mitochondrial respiratory chain complexes I and III in the SO muscle of the S-ND group, compared to that of the C-ND group (Figure 4). Kanazashi et al. reported that the SO muscle displays a decreased succinate dehydrogenase activity, an integral component of the mitochondrial respiratory chain, and increased oxidative stress during hindlimb suspension in rats [19]. Our previous study also demonstrates that mitochondrial dislocation during unloading conditions has deleterious effects on muscle fibers leading to atrophy and ROS leakage from the mitochondria [5]. The weakening of mitochondrial oxidative phosphorylation is usually accompanied with changes in mitochondrial biosynthesis. We found that the expression of energy and oxidative metabolism genes significantly decreased in the SO muscle of tail-suspension mice, while AX showed the regulating effect on the mitochondria biogenesis (Figure 6). Therefore, muscle atrophy stimulated by unloading stress in tail suspension is associated with the disturbance of mitochondrial ability including mitochondrial oxidative phosphorylation and mitochondrial biogenesis.

Numerous reports have proven that AX exerts its effects on mitochondria in fatty-liver disease, caused due to a high-fat diet, nonalcoholic steatohepatitis [38,39], gastric inflammation by oxidative stress [40], and cardiovascular diseases [41,42]. These unique characteristics of AX on mitochondria may relate to itself membrane structure. AX consists of conjugated double bonds and hydroxyl and keto groups that can embed in the cell membrane, from the inside to the outside. This feature confers strong antioxidant activity, which enables AX to react with the free radicals [43,44]. Our results indicated that AX was preferentially detected in the mitochondrial fraction and is consistent with previous reports of its accumulation in the mitochondria of normal human mesangial cells and blastocysts [15,30]. These findings strengthened the possibility that AX reacted to ROS produced from mitochondrial respiratory chain complexes, leading to the prevention of oxidative stress–related diseases, including muscle atrophy.

AnA is an inhibitor of the mitochondrial respiratory chain complex III, a major site of mitochondrial ROS generation, and strongly activates the production and release of superoxide anions into the inner mitochondria membrane space [45]. We examined the effect of AX on AnA-induced mitochondrial O^2-^ production using succinate as a complex II substrate. AX significantly suppressed mitochondrial complex III–driven ROS production in Sol8 myotubes (Figure 7a), whereas its effect was not observed in C2C12 myotubes (data not shown), which is likely to be involved in muscle fiber type. Sol8 and C2C12 cells were derived from SO and adult dystrophic mouse muscles, respectively. Indo et al. showed that the SO muscle, which is enriched with slow-twitch fibers, exhibits a higher production of ROS than fast-twitch fibers [3]. Some studies have also reported that dietary antioxidants reduce ROS production and ameliorate atrophy in the SO muscle more than other fast-twitch fibers [46,47,48]. These results indicate that AX could target the mitochondria to eliminate O^2-^ production and inhibit muscle atrophy induced due to mitochondrial oxidative stress in slow-twitch fibers.

Loss of MMP and excess ROS production in mitochondria leads to cytochrome c release from the mitochondria into the cytosol, resulting in the induction of apoptosis [49,50,51,52,53]. It has been revealed that overproduction of mitochondrial ROS, mitochondrial dysfunction, and mitochondria-mediated apoptosis play vital roles in skeletal muscle atrophy [54,55]. Caspase 3 is downstream of cytochrome c; the release of cytochrome c activates caspase 3, which induces apoptosis [56,57]. It has been reported that a deficiency in caspase 3 prevents denervation-induced muscle atrophy [58]. In addition, disturbed TUNEL-positive nuclei, increased caspase 3 protein level, and decreased Bcl-2, anti-apoptotic members that inhibit the release of cytochrome c by unloading were improved by AX [48]. In our present study, AX showed improvement of disturbed MMP as well as increased mitochondrial ROS by AnA treatment, thereby inactivating caspase 3 through an inhibition of cytochrome c release into cytosol in Sol8 myotubes. In agreement with these findings, AX has been shown to protect against decreased MMP by virtue of improving mitochondrial function in cancer and neural cells [14,59]. These results suggested that AX targeted and protected mitochondria by scavenging free oxygen radicals, regulating MMP, and inhibiting apoptosis in muscle cells.

## 5. Conclusions

In summary, the current study revealed that AX prevented muscle atrophy in slow-type muscles (SO). The direct effect of AX on mitochondria brought about the reduction of oxidative stress, regulation of mitochondrial function such as oxidative phosphorylation, biogenesis and MMP, and attenuation of apoptosis. These effects could collectively prevent the onset of muscle atrophy. Based on these results, AX could be considered as a potential treatment option for muscle atrophy and mitochondria-related diseases.

## Figures and Tables

**Figure 1 nutrients-13-00379-f001:**
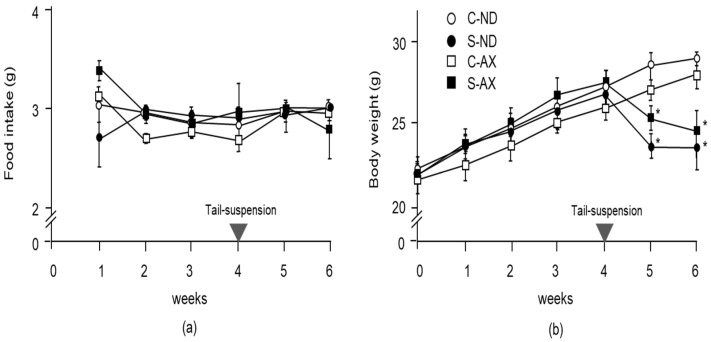
Changes in food intake and body weight with the astaxanthin (AX) diet: (**a**) food intake; (**b**) body weight. Mice were fed an AX or normal diet for 6 weeks. They were subjected to tail suspension at 4 weeks, which continued for 2 weeks (C-ND, *n* = 6; S-ND, *n* = 6; C-AX, *n* = 6; S-AX, *n* = 6). Data are presented as mean ± S.D. (*n* = 6). Statistical analysis was performed by the two-way ANOVA and Tukey test. * *p* < 0.05, compared with ND. C-ND, control mice fed the normal diet; S-ND, tail-suspension mice fed the normal diet; C-AX, control mice fed the AX diet; and S-AX, tail-suspension mice fed the AX diet. Arrow means the time of tail suspension.

**Figure 2 nutrients-13-00379-f002:**
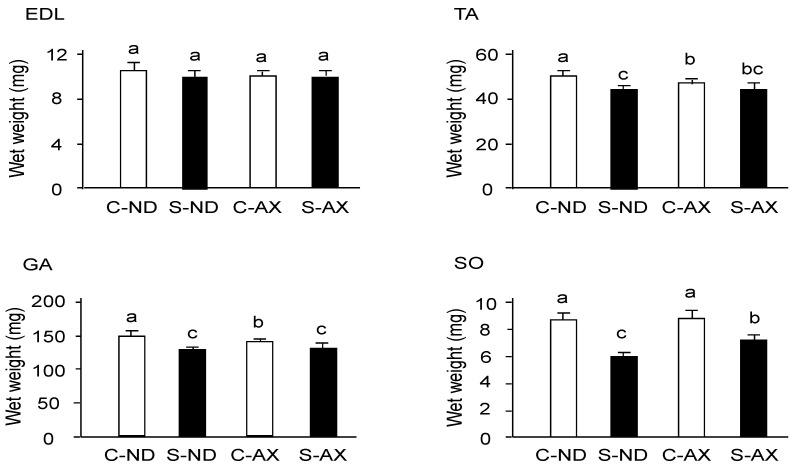
The effect of dietary AX on muscle mass in tail-suspension mice. An AX-supplemented or normal diet was given to the mice for 4 weeks and their skeletal muscles were isolated 2 weeks after tail suspension. The wet weights of the tibialis anterior, extensor digitorum longus, gastrocnemius, and soleus muscles were measured. Data are presented as mean ± S.D. (*n* = 6). Different letters indicate significant differences (*p* < 0.05) based on the two-way ANOVA and Tukey’s test. C-ND, control mice fed the normal diet; S-ND, tail-suspension mice fed the normal diet; C-AX, control mice fed the AX diet; and S-AX, tail-suspension mice fed the AX diet. TA, tibialis anterior muscle; EDL, extensor digitorum longus muscle; GA, gastrocnemius muscle, SO, soleus muscles.

**Figure 3 nutrients-13-00379-f003:**
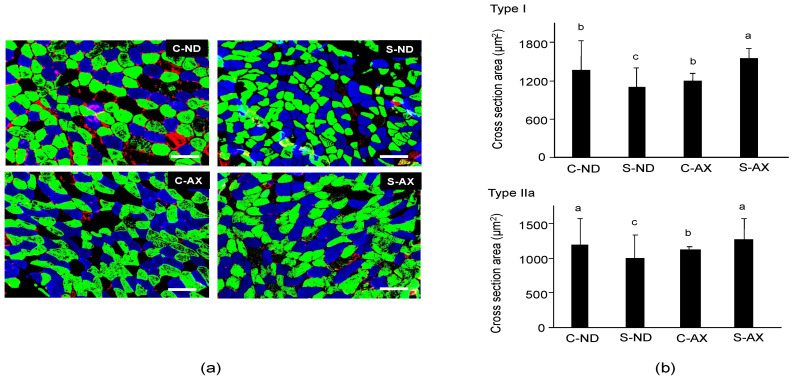
Effect of dietary AX on cross-sectional area (CSA) and fiber size in the soleus (SO) muscle of tail-suspension mice. (**a**) Sections (5 μm thickness) of SO muscle from C-ND, C-AX, S-ND, and S-AX groups, with multicolor immunofluorescence staining. Scale bar = 100 μm. Magnification, ×20; (**b**) The average CSA of type I and type IIa fibers in the soleus muscle. Data are presented as mean ± S.D. (*n* = 3). Different letters indicate significant differences (*p* < 0.05) based on the one-way ANOVA and Tukey’s test. C-ND, control mice fed the normal diet; S-ND, tail-suspension mice fed the normal diet; C-AX, control mice fed the AX-supplemented diet; and S-AX, tail-suspension mice fed the AX-supplemented diet.

**Figure 4 nutrients-13-00379-f004:**
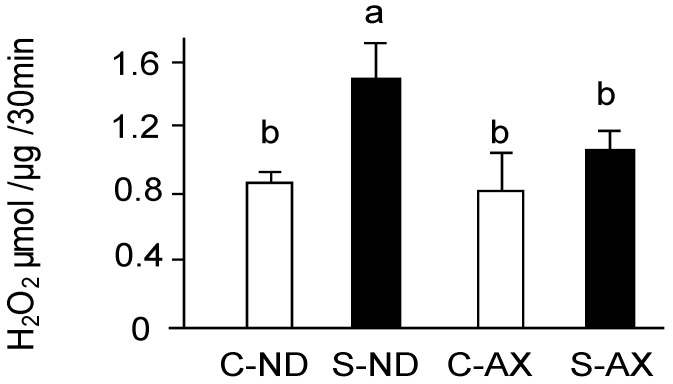
Effect of dietary AX on H_2_O_2_ production in the muscle of tail-suspension mice. The rate of H_2_O_2_ production from isolated mitochondria in muscle was detected by Amplex Red fluorescence. Data are represented as mean ± S.D. (*n* = 6). Different letters indicate significant differences (*p* < 0.05) based on the one-way ANOVA and Tukey’s test. C-ND, control mice fed the normal diet; S-ND, tail-suspension mice fed the normal diet; C-AX, control mice fed the AX-supplemented diet; and S-AX, tail-suspension mice fed the AX-supplemented diet. H_2_O_2_ production are representative of at least three independent studies.

**Figure 5 nutrients-13-00379-f005:**
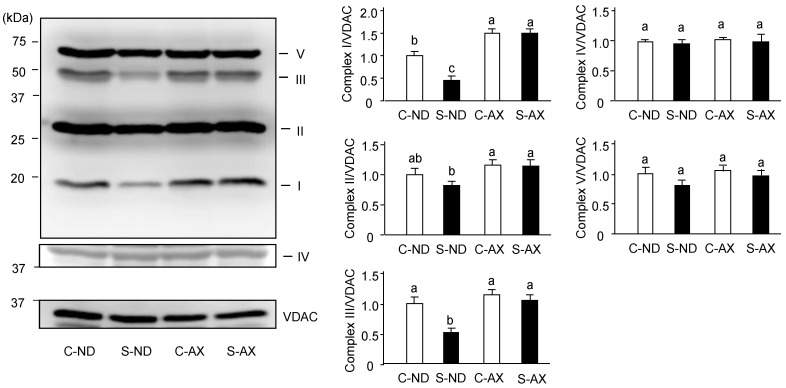
Effect of dietary AX on the expression of oxidative phosphorylation (OXPHOS)-related protein complexes in SO muscle of tail-suspension mice. Proteins (20 μg/lane) were extracted from the SO muscle and subjected to SDS-PAGE and transferred onto a PVDF membrane. Immunoblotting for total OXPHOS was performed. Data are represented as mean ± S.D. (*n* = 6). Different letters indicate significant differences (*p* < 0.05) based on the one-way ANOVA and Tukey’s test. C-ND, control mice fed the normal diet; S-ND, tail-suspension mice fed the normal diet; C-AX, control mice fed the AX-supplemented diet; and S-AX, tail-suspension mice fed the AX-supplemented diet. Immunoblotting experiments are representative of at least three independent studies.

**Figure 6 nutrients-13-00379-f006:**
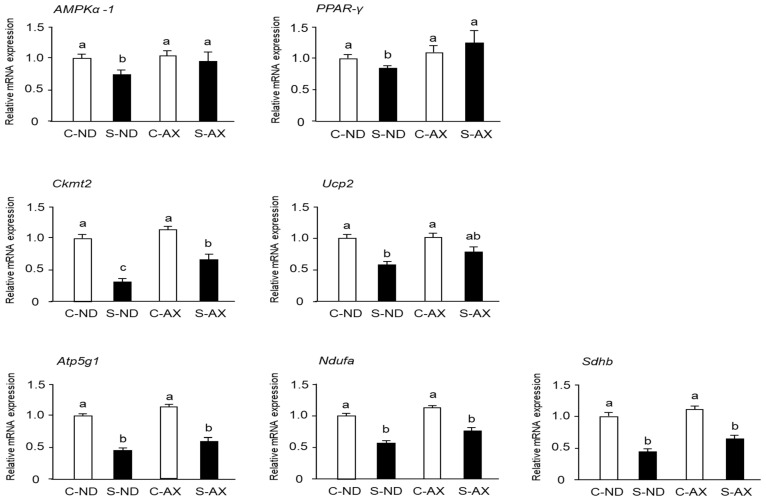
Effect of dietary AX on mitochondrial biogenesis in SO muscle of tail-suspension mice. The total RNA of the SO muscle was extracted and subjected to real-time PCR. Expression ratio relative to that of *glycelaldehyde-3-phosphate dehydrogenase* (*GAPDH*). Data are represented as mean ± S.D. (*n* = 6). Different letters indicate significant differences (*p* < 0.05) based on the one-way ANOVA and Tukey’s test. Real-time PCR experiments are representative of at least three independent studies. *AMPK*, *adenosine 5′-monophosphate (AMP)-activated protein kinase*; *PPAR*, *peroxisome proliferator-activated receptor*; *Ckmt*, *creatine kinase in mitochondrial*; *UCP*, *uncoupling protein*; *Atp5g1*, *ATP synthase*, *H+ transporting*, *mitochondrial F0 complex*, *subunit C1*; *Ndufaf2*, *NADH-ubiquinone oxidoreductase complex assembly factor*; *Sdhb*, *succinate dehydrogenase complex*, *subunit B*.

**Figure 7 nutrients-13-00379-f007:**
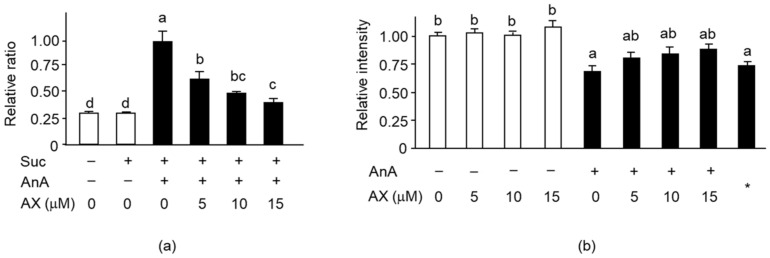
Effect of AX on mitochondrial superoxide levels and MMP in AnA-treated Sol8 myotubes: (**a**) The rate of superoxide formation in AX-treated Sol8 myotubes was assessed using dihydroethidium (DHE) fluorescence. Succinate and AnA were used as respiratory complex II substrate and complex III inhibitor, respectively; (**b**) The quantitative matrix metalloproteinase (MMP) values were calculated based on the ratio of fluorescence intensity values (JC-1 dye). Data are represented as mean ± S.D. (*n* = 8). Different letters indicate significant differences (*p* < 0.05) based on the one-way ANOVA and Tukey’s test. “−” represents DMSO; *, CCCP; Suc, succinate; AnA, antimycin A. ROS production and MMP levels experiments are representative of at least three independent studies.

**Figure 8 nutrients-13-00379-f008:**
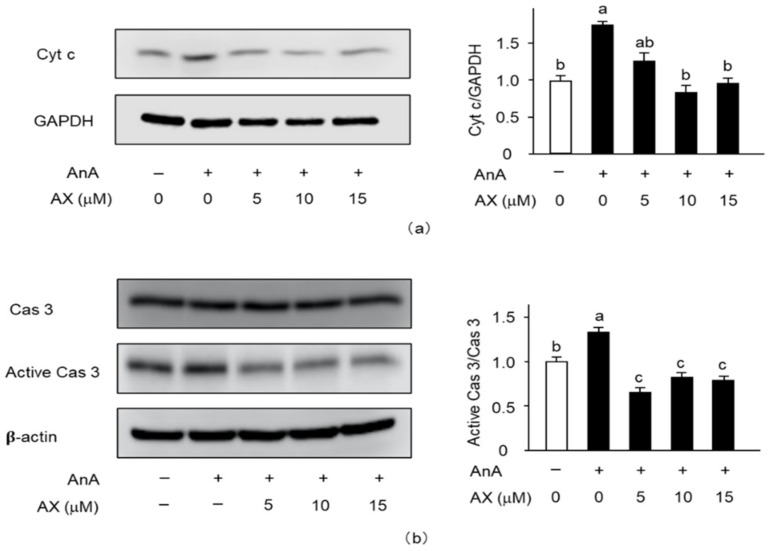
Effect of AX on the activationf apoptosis-related proteins in AnA-treated Sol8 myotubes: (**a**) Sol8 myotubes were treated with different AX concentrations for 1 h, then AnA (25 μM) was added for 48 h. The cytosolic fraction was subjected to SDS-PAGE and transferred onto a PVDF membrane. Immunoblotting for cyt c and GAPDH was performed on different membranes without antibody stripping, as described in Section 2.6. The ratio of cyt c proteins to GAPDH was calculated using densitometric analysis; (**b**) Sol8 myotubes were treated with different AX concentrations for 1 h and with AnA (25 μM) for 48 h before total proteins were extracted. Proteins (20 μg/lane) were extracted from Sol8 myotubes and subjected to SDS-PAGE, then transferred onto a PVDF membrane. Immunoblotting for cyt c, cas 3, and active cas 3 was performed on different membranes without antibody stripping, as described in Section 2.6. Data are presented as mean ± S.D. (*n* = 3). Different letters indicate significant differences (*p* < 0.05) based on the one-way ANOVA and Tukey’s test. AnA, antimycin A; cyt c, cytochrome c; and cas 3, caspase 3. Immunoblotting experiments are representative of at least three independent studies.

**Table 1 nutrients-13-00379-t001:** AX content in mitochondria and cytosol of AX-treated Sol8 myotubes.

AX Treatment	AX Content (nmol)	
	Cytosol	Mitochondria
0 nmol	N.D.	N.D.
100 nmol	0.09 ± 0.01	1.07 ± 0.02

Data are presented as mean ± S.D. (*n* = 3). N.D.: not detected; AX: Astaxanthin.

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
