# Peer review of "Astaxanthin Prevents Atrophy in Slow Muscle Fibers by Inhibiting Mitochondrial Reactive Oxygen Species via a Mitochondria-Mediated Apoptosis Pathway"

_nutrients, 2021, doi:10.3390/nu13020379_

Round 1

Reviewer 1 Report

Remarks to the Authors

The purpose of the current study was to determine the role of astaxanthin supplementation in preventing muscle atrophy in a hind-limb suspension model.

This reviewer commends the authors on their work investigating the protective effects of astaxanthin supplementation in response to disuse muscle atrophy. However, several studies have previously investigated the effects of astaxanthin supplementation with disuse muscle atrophy and have made many of the measurements assessed in the current findings. Due to the unpredictable nature of disuse-atrophy events, supplementing astaxanthin four weeks prior to a disuse atrophy event would not likely be a viable treatment option. Given the amount of prior work investigating astaxanthin supplementation, studies investigating a shorter time-frame of supplementation would seem to have a higher impact at this time. Additionally, the manuscript would be improved by the authors addressing several methodological questions. Major and minor remarks follow.  

Major comments

Line 122- How was data for DHE normalized? If using a micro-plate reader, how did the authors ensure that equal amounts of cells were present between groups? DHE also undergoes auto-oxidation, at what time-point were the respective measures made? Additionally, antimycin has been shown to react with DHE and increases its fluorescence (Tollefson 2003). Did the authors take this into consideration when making this measurement? Lastly, the added source does not provide any additional information and should be removed.

Line 182- The authors say that CSA was analyzed, but were any of these values compared statistically? Figure 3 currently depicts quantification of these values but does not indicate statistical analysis. Additionally, why was fiber typing performed if these values were going to be displayed in sub-groups of CSA?

Line 224- Why was AX accumulation data measured using 100pmol when other measures were made with 150,000x higher concentrations (i.e., 15μM in DHE experiments vs 100pmol in accumulation)?

Figure 2- It appears that AX supplementation decreased the weights of gastroc and tibialis anterior muscles in control animals. Can the authors comment on this?

Figure 4- Where is the quantification of these values? This is a nice representative image, however the sample size and bar graph need to be displayed. Additionally, what was the loading control for this measure? These points need to be addressed in order to say that these measures were decreased.

Minor comments-

Line 27- currently reads AX decreased muscle soleus fiber size. Revise for clarity

Author Response

 Point-by-point response to the comments of Reviewer 1

We thank the reviewer for evaluating our manuscript. Thanks for the great suggestions and comments. As for the suggestion of “shorter time-frame of supplementation”, We will gradually adjust in future research to achieve a better model. Based on the comments, we conducted new experiments, provided more details and modified/added the figures. The following text describes our response to the comments made by the reviewer. All line numbers mentioned in each response to each comment refer to the numbers that appear on the left margin of the text of the revised manuscript.

Major comments

  1. Line 122- How was data for DHE normalized? If using a micro-plate reader, how did the authors ensure that equal amounts of cells were present between groups? DHE also undergoes auto-oxidation, at what time-point were the respective measures made? Additionally, antimycin has been shown to react with DHE and increases its fluorescence (Tollefson 2003). Did the authors take this into consideration when making this measurement? Lastly, the added source does not provide any additional information and should be removed.

Response:     

Thanks for your helpful comment. Based on your comment, we have improved the manuscript and data of Figure 7a.

Calibration with the equal amounts of cells:

To measure the amounts of cells, cell viability was examined using the Cell Counting Kit-8 (CCK-8) (Dojindo, Kumamoto, Japan). Briefly, the Sol8 cells (3×103/well) were seeded into clear and black 96 well-plate for measurement of cell viability and O2- production, respectively. After 3 days, Sol8 myoblasts were fused by shifting the medium to DMEM supplemented with 2% HS. Cells were maintained in 2% HS/DMEM (differentiation medium) for 4 days and were treated with AX (final concentration: 0, 5, 10, 15 μM) for 24h. To measure the cell number, the cultured medium (supernatant) was removed and washed by HBSS for 3 times. Then, treated CCK-8 (10 μL /100 μL HBSS) in each well for 30min, absorbance at the wavelength of 450 nm was read using a microplate reader. The result showed that there were no significant differences among all groups in the cell viability. As shown in below, there were no significant differences among all groups. Therefore, we continued the experiment.

DHE normalized:

As mentioned in the previous report (Tollefson 2003), more than 10 μg/mL concentration of antimycin A reacts with DHE and increase the fluorescent slightly. In our studies, we used 10 μM antimycin A (=approximately 5.5 μg / mL) for incubation 30min.To eliminate these factors, we normalized by blank (buffer containing DHE plus antimycin A), and normalized data showed as below. In addition, we have explained the measurement of ROS using DHE dye in Material and Methods (Page 4, lines 148-151). This data was modified to the Results (Page 9, Fig.7a).

  1. Line 182- The authors say that CSA was analyzed, but were any of these values compared statistically? Figure 3 currently depicts quantification of these values but does not indicate statistical analysis. Additionally, why was fiber typing performed if these values were going to be displayed in sub-groups of CSA?

Response:     

Thanks you for your helpful comment. Because other reviewers also suggested that we need statistics data of CSA, we modified the data of CSA from the line graph of the muscle fiber distribution to the histogram after statistical analysis of the data (showed in Result Fig.3b). The reduction in muscle CSA caused by tail suspension usually occurs in slow-type muscles such as SO, the muscle atrophy of mice is usually accompanied by the transformation of different muscle fiber types in the muscle. Based on the result of muscle weight in Fig.2, we found AX showed resistant to the decrease of SO muscle. To determine which type of muscle fiber in SO influenced by AX, we analyzed the distribution of muscle fiber distribution. Consistent with the results that indicated a decrease in muscle weights in the S-ND group, the average CSA of type I and IIa muscle fibers in the S-ND group was significantly decreased, compared to that in the C-ND group. Meanwhile, AX prevented the decrease of CSA caused by tail-suspension and resulted in the increase of CSA rather than the reduction of CSA (Fig.3b). These information added to the manuscript (Page 5 and 6, lines 220-224).

Fiber typing:

It has been reported that tail-suspension affect slow fiber type muscle, especially type I and IIa, but not fast fiber type (Ohira T et al., 2014). Because the soleus muscle consists with type I fiber (30.6 ± 2.2%), type IIa fiber (49.1 ± 1.2%), type IIx (11.8 ± 1.7%) and other type of fibers, we performed different fiber typing to investigate which type fiber influenced by AX (Bloemberg D, et. Al., 2012).

These information added to the manuscript (Page 5, lines 212-216).

  1. Line 224- Why was AX accumulation data measured using 100pmol when other measures were made with 150,000x higher concentrations (i.e., 15μM in DHE experiments vs 100pmol in accumulation)?

Response:     

Dear reviewer, we must apologize for this. This is a mistake. We have changed it to “100 nmol” (Page 3, lines 129-130; Page 9, Table 1). In addition, we explained the detail of method (2.5 detection of AX in mitochondrial and Cytosolic Fractions): Briefly, Sol8 myotubes were treated with AX (100 nmol) or DMSO, as control, in 10 mL culture medium/100 mm dishes and incubated at 37°C with 5% CO2 for 24 h. (Page 3, lines 129-130).

  1. Figure 2- It appears that AX supplementation decreased the weights of gastroc and tibialis anterior muscles in control animals. Can the authors comment on this?

Response:     

Although GA and TA are composed of most fast-twitch fibers, they also contain intermediate muscle fibers such as IIa (GA: 20.9 ± 1.6%, TA: 18.2 ± 2.4%) (Bloemberg 2012). In contrast, EDL is occupied by fast-twitch fibers such as IIb. Indeed, we found that the size of IIa muscle fibers in the S-ND group showed significantly decreased, compared with the C-ND group (Figure 3b). Thus, muscle atrophy caused by tail suspension preferentially affected slow-type rather than fast-type muscles. Moreover, AX prevented atrophy in the muscle containing type I and IIa myofibers. These findings suggest that AX acts in slow-switch and intermediate muscle fibers. This information is added to the Discussion (Page 10 and 11, lines 367-373).

  1. Figure 4- Where is the quantification of these values? This is a nice representative image, however the sample size and bar graph need to be displayed. Additionally, what was the loading control for this measure? These points need to be addressed in order to say that these measures were decreased.

 Response:    

Thanks to your helpful comment. We have added the VDAC western blotting data for loading control, and calculated the ratio of OXPHOS to VDAC. We found AX prevented the increase of H2O2 production caused by tail-suspension. Thus, we investigated the effect of AX on the levels of oxidative phosphorylation-related protein complexes in the muscle. The amounts of complexes I and III in the SO muscle of the S-ND group were significantly lower than that of the C-ND group. In contrast, there were no significant changes between the protein complexes in the C-AX and S-AX groups (Fig.5)

This information is added to the Result (Page 7, lines 263-267 and Fig.5).

Minor comments-

Line 27- currently reads AX decreased muscle soleus fiber size. Revise for clarity

Response:     

Thanks to your helpful comment. We have modified it (Page 1, lines 27-28).

Reviewer 2 Report

The manuscript describes the interesting phenomenon that diet containing Astaxanthin (AX) prevented slow fiber muscle atrophy in mice caused by tail suspension (inactivity). The study attributed this muscle atrophy prevention of AX to its inhibition of mitochondrial ROS and consequential mitochondria-mediated apoptosis. Unfortunately, most of these data are performed in Sol8 cells without direct in vitro-in vivo correlation, neither with direct evidence showing its effects on apoptosis.

It seems appropriate that there would be further requirements of mechanistic analysis, such as what’s the direct target of AX working on mitochondria and what’s the inhibitory mechanism of AX on oxidative phosphorylation complex II and III (at transcription or protein level?) and is there any known signaling pathway may be involved, etc.

Specific comments:

  1. Figure 4: Please add a loading control to the western blot results. In addition, it seems complex IV and V are over-saturated. It would be appropriate to detect complex IV and V at different exposure from complex II and III, if the intensity of these proteins are greatly different.

        Additionally, it would be helpful to measure these proteins in fast muscles (i.g.: EDL) in parallel to show the selectivity of AX effect on slow muscles.

  1. Figure 6: it would make the argument stronger if there could be additional data supporting the AX-inhibited apoptosis, such as TUNEL staining or Hoechst/PI staining.
  2. Besides, it would better translate the in vitro results to in vivo by measuring Cyt C level, Cas3 activity and/or apoptosis in vivo (SO muscle) in normal and tail suspension animals with or without AX-treatment.

Author Response

 Point-by-point response to the comments of Reviewer 2

We thank the reviewer for evaluating our manuscript. Thanks for the great comments and suggestions. Due to the limitation of the amount of SO muscle in mice, we failed to complete the further mechanism analysis and other signaling pathway. We feel sorry for that, and we will try experiment for further mechanism analysis. Based on the comments, we modified the manuscript. The following text describes our response to the comments made by the reviewer. All line numbers mentioned in each response to each comment refer to the numbers that appear on the left margin of the text of the revised manuscript.

Specific comments:

  1. Figure 4: Please add a loading control to the western blot results. In addition, it seems complex IV and V are over-saturated. It would be appropriate to detect complex IV and V at different exposure from complex II and III, if the intensity of these proteins are greatly different.

        Additionally, it would be helpful to measure these proteins in fast muscles (i.g.: EDL) in parallel to show the selectivity of AX effect on slow muscles.

Response:

We added VDAC as inner control, and the expression of complex were re-analyzed by Image J software (National Institutes of Health, Bethesda, MD, USA). This information was modified in Materials and Methods, and Results (Page 4, lines 180 and 184-185, Page 7, Fig.5).

OXPHOS in EDL:

We have done the western blotting of OXPHOS in EDL fast muscle. As shown in below, we found suspended EDL muscle fail to decrease the amount of OXPHOS proteins compare to control. However, the complex I protein in AX diet group was slightly decreased, compared with that in normal diet group. These findings were contained some controversial result. To elucidate this mechanism, we need further examination. Therefore, we did not use this data in this manuscript.

  1. Figure 6: it would make the argument stronger if there could be additional data supporting the AX-inhibited apoptosis, such as TUNEL staining or Hoechst/PI staining.

Response:

Previous report showed the number of TUNEL-positive nuclei per section of soleus muscle fibers increased significantly in the hindlimb unloading groups, compared to the normal group. (Yoshihara 2017, photo shown as below). However, this increase in TUNEL-positive nuclei was significantly suppressed in the hindlimb unloading with AX groups. This result also indicated the AX inhibited apoptosis.

This information added to the discussion (Page12, lines 444-446).

  1. Besides, it would better translate the in vitro results to in vivo by measuring Cyt C level, Cas3 activity and/or apoptosis in vivo (SO muscle) in normal and tail suspension animals with or without AX-treatment.

Response:

Numerous studies have shown that the apoptosis will occur under the hindlimb unloading condition. However, the ROS-mediated apoptosis prevented in the rat soleus muscle during hindlimb unloading due to the ROS scavenging effects of AX supplementation. The increased level of caspase 3 and decreased Bcl-2, anti-apoptotic members that inhibit the release of cytochrome c, by unloading was improved by AX. (Yoshihara 2017, photo shown as below).

This information added to the discussion (Page12, lines 444-446).

Reviewer 3 Report

Sun and colleagues aimed to examine the effects of Astaxanthine (AX) on skeletal muscle mass during hindlimb suspension in young male mice. The authors propose that AX prevents muscle mass loss through mitigating muscle oxidative stress; however, there are substantial concerns regarding the statistics run throughout the experiment and the weak link between in vivo results and in vitro mechanisms. There is no evidence of increased ROS in hindlimb suspenden mice and there is no evidence of atrophy in vitro. The in vitro results are intriguing and suggest AX can mitigate oxidative stress; however, these finding alone are not sufficient for publication.  

Specific Comments 

Introduction

There is a cursory mention of the different atrophic conditions and the mechanisms that contribute to disuse-induced atrophy. This needs additional attention in the introduction.

 Methods

 Animals

Male mice achieve sexual maturity around 8-10 weeks. There is no justification for using adolescent mice (6 wks of age).

How much AX was ingested for each mouse? Total food intake for each group does not shed light on the consistency of the drug deliver across the AX group.

 IF staining

How were the sections cut and prepared? It is not clear if you are cutting through the mid belly of the Soleus. Additionally, what magnification were the pictures take at?

There is no justification for only measuring 600 fibers per group? What is the inter/intra-animal variability? You need to measure enough fibers to stabilize the Standard Deviation within each mouse.

Cell culture

4 days of differentiation is not sufficient, likely assessing myoblasts still. Can you validate the presence and/or absence of myoblasts?

How many replicated were run for each experiment? It is standard to run 3 separate experiments with 3 replicates within those experiments.

Statistics

The statistics are inadequate for the completed experiments. In order to appropriately determine an effect of a treatment (AX) you need to use a two-way ANOVA. It is unlikely then that your difference in muscle weights would achieve statistical significance. This is extremely concerning given that there are no mechanisms addressed in vivo and you are relying on this data for rational to examine mechanisms in vitro.

It is unclear what the N was  of each experiment. Either provide that information in the figure legends for each figure or in the methods with more detail.

Results

You need to run a repeated measures ANOVA for Figure 1A+B and a Two-Way ANOVA for the final body weights and undoubtably you will achieve a main effect of hindlimb suspension on body weight.

Please indicate visually when the hindlimb suspension started on Figure 1A+B

Discuss the relative food intake / body weight ratio at 5 and 6 weeks.

Figure 2 (TA) indicate that AX exacerbated TA muscle weight loss, please discuss.

Histology is not convincing, the fluorescence is spotty and there is red staining throughout the extracellular matrix suggesting significant background and nonspecific staining. Additionally, what are the black unstained fibers?

What is the mean CSA? This would help bolster the muscle weight data.

It is not clear what the authors mean by “was almost the same”. What does this mean?

Were there any statistics run on CSA and fiber type abundance?

Did you quantify the results from the OXPHOS blot? Also where where is Complex I? This blot needs a loading control (VDAC)? What was the N of this particular experiment?

Cell culture data is insufficient to support the mechanistic claims proposed.

Discussion

There is no evidence to suggest AX prevented muscle mass loss through mitigating ROS. There is no evidence that ROS is increased in their hindlimb suspended mice nor is there evidence that AX reduced this alleged increase in ROS. The atrophic mechanisms proposed in vitro are independent may be independent of hindlimb suspension.

Author Response

Point-by-point response to the comments of Reviewer 3

We thank the reviewer for evaluating our manuscript. Based on the comments, we conducted new experiments, provided more details and modified/added the figures. The following text describes our response to the comments made by the reviewer. All line numbers mentioned in each response to each comment refer to the numbers that appear on the left margin of the text of the revised manuscript.

Comments and Suggestions:

There are substantial concerns regarding the statistics run throughout the experiment and the weak link between in vivo results and in vitro mechanisms.

There is no evidence of increased ROS in hindlimb suspenden mice and there is no evidence of atrophy in vitro.

Response:

Numerous studies have demonstrated that mitochondria play an important role in muscle atrophy (Nikawa T, 2004, FASEB. J.; Adhihetty PJ, 2005, Am. J. Physiol. Cell Physiol.; Marzetti E, 2010, Biochem. Biophys. Acta.) Consistent with these findings, we found that the amount of H2O2 production caused by tail-suspension was significantly increased by tail-suspension in normal diet mice (S-ND group). However, AX-supplemented diet inhibited the increase of H2O2 production in S-AX group.

This information and new data added in the manuscript (Page 7, Lines 245-250; Fig.4).

Mitochondrial respiratory chain complexes I and III are major sites of ROS leakage. To confirm this result, we continued to investigate the effect of AX on the OXPHOS complex expression. The amounts of complexes I and III in the SO muscle of the S-ND group were significantly lower than that of the C-ND group. In contrast, there were no significant changes between the protein complexes in the C-AX and S-AX groups. This information added in the manuscript (Page 7, Lines 263-267, Fig.5)

Oxidative stress induced by mitochondrial dysfunction plays an important role in muscle atrophy. Therefore, we used antimycin A, an inhibitor of complex III, to make a model of mitochondrial dysfunction in vitro experiment.

Specific Comments

  1. Introduction

There is a cursory mention of the different atrophic conditions and the mechanisms that contribute to disuse-induced atrophy. This needs additional attention in the introduction.

Response:

Mitochondria play a vital role in disused skeletal muscle atrophy (Nikawa T, 2004) and mitochondrial signaling can contribute to disuse muscle atrophy in 3 major ways (Powers SK, 2011; Romanello V, 2010; Adhihetty PL, 2005): (1) increased mitochondrial ROS production, (2) energy stress: decrease in ATP production and the activation of AMPK, (3) mitochondria release of pro-apoptotic factors: release of cytochrome c into the cytosol and active caspase 3. In fact, the increase of mitochondrial ROS production could participate in both the energy stress and apoptosis (Kowaltowski AJ, 1999). Therefore, mitochondrial ROS have major impact on disuse muscle atrophy and the development of antioxidants to prevent the increase in mitochondrial ROS in muscle due to inactivity has been extensively researched.

This information added in the manuscript (Page 2, lines 50-56).

  1. Methods

2-1 Animals

Male mice achieve sexual maturity around 8-10 weeks. There is no justification for using adolescent mice (6 wks of age).

Response:

As shown in previous studies, 7-8 weeks old C57BL/6J mice were used before randomization in disuse muscle atrophy model (Mukai R, 2016, Am J Physiol Regul Integr Comp Physiol; Tomiya S, 2019, Am J Physiol Regul Integr Comp Physiol). Consist with previous studies, we have planned as below figure. Briefly, after 1 week of acclimation feeding, mice were randomized into 4 groups (C-ND, S-ND, C-AX, S-AX). Mice were fed with AX-supplemented diet (C-AX group and S-AX group) or normal diet (C-ND group and S-ND group) for 4 weeks. Since then, these mature mice were up to 11 weeks, they were continued to fed with AX-supplemented diet (C-AX group and S-AX group) or normal diet (C-ND group and S-ND group) for 2 weeks, at the same time S-AX group and S-ND group were subjected to tail-suspension to establish the muscle atrophy model.

This information added to the manuscript (Page 2, Lines 81-85).

How much AX was ingested for each mouse? Total food intake for each group does not shed light on the consistency of the drug deliver across the AX group.

Response:

After calculated the food intake data, we found that the amounts of AX, 0.55 ± 0.15 mg / day were fed to each mouse. Although astaxanthin has low solubility ranges from 10 to 50% (Nagao A, 2009), supplemented-AX administration had several functions in tissues (Kanazashi M, 2014, Exp Phosiol; Yoshihara T, 2017, J Physio Sci). Given these findings, AX has potent action even at low concentration in tissues.

This information added to the manuscript (Page 5, Lines 194-195).

2-2  IF staining

How were the sections cut and prepared? It is not clear if you are cutting through the mid belly of the Soleus. Additionally, what magnification were the pictures take at?

Response:

The separated SO muscle was placed on the gum tragacanth and immediately frozen in chilled isopentane with liquid nitrogen and stored at -80°C until analysis. The mid-belly transverse cryosections of SO muscles (5-μm thickness) were made with LEICA CM1950 cryostat (Wetzlar, Germany). Sections were put on poly-L-lysine coated slide glasses.  Magnification: 20-fold.

These information added in the manuscript (Page 3, lines 101-105; Page 6, Fig.3, line 239).

There is no justification for only measuring 600 fibers per group? What is the inter/intra-animal variability? You need to measure enough fibers to stabilize the Standard Deviation within each mouse.

Response:

SO muscles (n=3) were used in IF staining, and at least 600 myofiber cross-sectional areas (CSAs) were measured per group. In addition, we re-calculate the average CSA of types I and IIa fibers in the SO muscle and two-way analysis of variance, using SPSS statistics software, followed by the Tukey test for individual differences between groups. Data were represented as mean ± S.D. (n=3). Consistent with the results that indicated a decrease in muscle weights in the S-ND group, the average CSA of type I and IIa muscle fibers in the S-ND group was significantly decreased, compared to that in the C-ND group. Meanwhile, AX prevented the decrease of CSA caused by tail-suspension and resulted in the increase of CSA rather than the reduction of CSA (Fig. 3b).

This information added and modified in the manuscript (Page 5, lines 220-224, Page 6, lines 240-241).

2-3 Cell culture

4 days of differentiation is not sufficient, likely assessing myoblasts still. Can you validate the presence and/or absence of myoblasts?

Response:

As shown in below figure, we found fused myofibers after 4days differentiation of sol8. Moreover, we can detecte slow type myosin heavy chain in sol8 myotubes. Consistent with this result, myogenine, muscle differentiation marker, can be detected in sol8 myotubes after differentiation 36 hours, and degree of myotube fusion cultured by differentiation media for 96 hours was around 30% (Kim MJ, 2010, PLoS ONE). Thus, we used 4 days of differentiation in this study.

How many replicated were run for each experiment? It is standard to run 3 separate experiments with 3 replicates within those experiments.

Response:

All experiments data are representative of at least three independent studies.

This information added to the manuscript (Page 7, Lines 259-260; Page 8, Lines 277-278; Page 8, Lines 295-296; Page 9, Lines 330-331; Page 10, Lines 353-354).

2-4  Statistics

The statistics are inadequate for the completed experiments. In order to appropriately determine an effect of a treatment (AX) you need to use a two-way ANOVA. It is unlikely then that your difference in muscle weights would achieve statistical significance. This is extremely concerning given that there are no mechanisms addressed in vivo and you are relying on this data for rational to examine mechanisms in vitro.

Response:

We agree with your comments. To achieve statistical significance, all data were re-analyzed by one / two-way analysis of variance, using SPSS statistics software, followed by the Tukey test for individual differences between groups. P-values < 0.05 were considered to indicate significant differences. This information added in the manuscript (Page 5, Lines187-189).

It is unclear what the N was of each experiment. Either provide that information in the figure legends for each figure or in the methods with more detail.

Response:

We added these information in every figure legends of the manuscript.

  1. Results

3-1  You need to run a repeated measures ANOVA for Figure 1A+B and a Two-Way ANOVA for the final body weights and undoubtably you will achieve a main effect of hindlimb suspension on body weight.

Response:

Statistical analysis was performed by the two-way ANOVA and Tukey test. *P < 0.05, compared with ND.

This information added in the manuscript (Page 5, lines 203-205, Fig. 1b).

3-2  Please indicate visually when the hindlimb suspension started on Figure 1A+B

Response:

We modified in the manuscript (Page 5, Fig.1).

3-3  Discuss the relative food intake / body weight ratio at 5 and 6 weeks.

Response:

We found that the ratio of body weight / food intake decreased significantly at 5 and 6 weeks in tail-suspension group (S-ND and S-AX). Previous studies showed that there was an increased reliance on carbohydrate metabolism for energy associated with muscle unloading (Stein TP, 2005, J Nutri Biochem). In addition, shift of fiber phenotype was related to downregulation of mitochondrial proteins and upregulation of glycolytic protein, suggesting a shift from oxidative to glycolytic metabolism (Ohira T, 2014, Physiological reports). Consistent with these results, we found that the decreased ratio of body weight / food intake and the shift of type fibers caused by tail-suspension. Additionally, we found that the mRNA expression of oxidative metabolism related transcription factors was decreased in tail-suspension group. Thus, these results showed that the decreased ratio of body weight / food intake associated with the energy shift from oxidative to glycolytic metabolism.

This information added in the manuscript (Page 11, lines 386-395). 

3-4  Figure 2 (TA) indicate that AX exacerbated TA muscle weight loss, please discuss.

 Histology is not convincing, the fluorescence is spotty and there is red staining throughout the extracellular matrix suggesting significant background and nonspecific staining. Additionally, what are the black unstained fibers?

Response:

The CSA of type I and IIa myofibers in S-AX group significantly increased, compared with S-CN or other groups (Figure 3). SO muscle contains type IIx (11.8 ± 1.7%) and type IIb (3.1 ± 1.1%) as well as type I (30.6 ± 1.2%), type IIa (49.1 ± 1.2%) (Bloemberg D, 2012, ref. 21). Given the influence of AX in type I and IIa muscle fibers, AX is likely to contribute to the transformation of type IIx to type I and IIa in SO muscle of tail-suspension mice, indicating increased CSA in S-AX. Likewise, TA muscle contains type IIx (44.7 ± 11.9%, CSA: 2186.0 ± 35.2 μm2), type I (0.6 ± 1.6%, CSA: 1501.4 ± 7.1 μm2) and type IIa (18.2 ± 2.4%, CSA: 1369.6 ± 22.4 μm2). Additionally, GA muscle contains type IIx (19.6 ± 2.1 %, CSA: 2186.0 ± 35.2 μm2), type I (7.9 ± 0.5%, CSA: 1743.4 ± 28.2 μm2) and type IIa (41.6 ± 1.3%, CSA: 1346.2 ± 22.8 μm2). We found that the wet weight of TA and GA in C-AX group was significantly decreased, compared with C-ND group. AX is presumed to be the transformation of type IIx to type I and IIa muscle fibers, resulting in the shift to small size fibers. However, further studies are necessary to define this mechanism.

Red staining represented the type IIb myofibers (we showed in manuscript). Although slow type muscle fiber such as type I and IIa are mainly exist in soleus muscle, there also have some other fiber type such as IIb and IIx in the soleus muscle (type I, 35%; type IIa, 53%; type IIX, 11%; type IIb, 1%. Komiya Y, 2017). Black unstained fibers are represented as type IIx fiber.

These information added to the manuscript (Page 11, lines 373-384).

3-6  What is the mean CSA? This would help bolster the muscle weight data.

 It is not clear what the authors mean by “was almost the same”. What does this mean?

 Were there any statistics run on CSA and fiber type abundance?

Response:

We modified the data of CSA from the line graph of the muscle fiber distribution to the histogram after statistical analysis of the data (showed in Result Fig.3b). The reduction in muscle CSA caused by tail suspension usually occurs in slow-type muscles such as SO, the muscle atrophy of mice is usually accompanied by the transformation of different muscle fiber types in the muscle. Based on the result of muscle weight in Fig.2, we found AX showed resistant to the decrease of SO muscle. To determine which type of muscle fiber in SO influenced by AX, we analyzed the distribution of muscle fiber distribution. Consistent with the results that indicated a decrease in muscle weights in the S-ND group, the average CSA of type I and IIa muscle fibers in the S-ND group was significantly decreased, compared to that in the C-ND group. Meanwhile, AX prevented the decrease of CSA caused by tail-suspension and resulted in the increase of CSA rather than the reduction of CSA (Fig.3b).

These information added to the manuscript (Page 5 and 6, lines 212-216 and 220-225).

3-9  Did you quantify the results from the OXPHOS blot? Also where where is Complex I? This blot needs a loading control (VDAC)? What was the N of this particular experiment?

Response:

We added VDAC as inner control, and the expression of complex were re-analyzed by Image J software (National Institutes of Health, Bethesda, MD, USA). This information was modified in Materials and Methods, and Results (Page 4, lines 180-181 and 184-185, Page 7, Fig.5).

3-10  Cell culture data is insufficient to support the mechanistic claims proposed. 

Response:

Numerous studies have shown that the apoptosis will occur under the hindlimb unloading condition. However, the ROS-mediated apoptosis prevented in the rat soleus muscle during hindlimb unloading due to the ROS scavenging effects of AX supplementation. The increased level of caspase 3 and decreased Bcl-2, anti-apoptotic members that inhibit the release of cytochrome c, by unloading was significantly controlled by AX. (Yoshihara 2017, photo shown as below). To confirm these findings, we performed AX treatment experiment by using sol8 myotubes.

This information added to the discussion (Page12, lines 444-446).

  1. Discussion

 There is no evidence to suggest AX prevented muscle mass loss through mitigating ROS. There is no evidence that ROS is increased in their hindlimb suspended mice nor is there evidence that AX reduced this alleged increase in ROS. The atrophic mechanisms proposed in vitro are independent may be independent of hindlimb suspension.

 Response:      

We conducted new experiments, provided more details and modified/added the figures and discussion section. In this study, the rate of H2O2 production by isolated mitochondria was detected using a fluorescent probe, Amplex Red. Mitochondria from muscle tissues were suspended in a buffer containing 5 mM MOPS (pH 7.4), 70 mM sucrose, and 220 mM mannitol, and mitochondria protein concentration was determined by using a BCA protein assay (Pierce, Rockford, IL, USA). Mitochondria (30 μg mitochondrial protein per well) were incubated in black 96 well-plate with a reaction mixture containing 50 μM Amplex Red, 2 U/ml horseradish peroxidase, 30 U/ml superoxide dismutase (SOD), 10 mM succinate, 10 μM antimycin A as substrate at room temperature for 30 min, protected light. SOD was added to convert all superoxide into H2O2. SOD was added to convert all superoxide into H2O2. The rate of H2O2 production was linear with respect to mg of mitochondrial protein.

we found that the amount of H2O2 production caused by tail-suspension was significantly increased by tail-suspension in normal diet mice (S-ND group). However, AX-supplemented diet inhibited the increase of H2O2 production in S-AX group.

These information added in the manuscript (Page 3, Lines 121-126, 136-146; Page 7, Lines 245-260; Page 11, Lines 399-402).

In addition, to further explore the mechanism of AX in the SO muscle mitochondria of tail-suspension mice. we conducted new experiment, we take mRNA from SO muscle and investigate the effect of AX on mitochondria biogenesis by real time RT-PCR.

 When muscle under immobilization and denervation, a series of undesirable changes occur in the mitochondria of the muscle, including inhibition of mitochondrial biogenesis (ref. 27). Moreover, it has been reported that AX treatment stimulates mitochondrial biogenesis in the skeletal muscle (ref. 28). Consist with previous reports, we found that AMPKα-1 and PPAR-γ mRNA expressions in S-ND group were significantly decreased, compared with these in C-ND group. Whereas the S-AX group showed inhibition of the decrease of these mRNA expressions in SO muscle. In addition, the expression of Ckmt2 mRNA in S-AX group was significantly higher than that in S-ND group. Although there were no significantly differences between S-ND and S-AX in the expression of Ucp2, Atp5g1, Ndufa and Sdhb, AX improved the expression slightly in tail-suspension mice. (Figure 6).

AX promoted mitochondrial biogenesis by up-regulating the expressions of Adenosine 5‘-monophosphate-activated protein kinase (AMPK) α-1, peroxisome proliferator-activated receptor (PPAR)-γ, creatine kinase in mitochondrial (Ckmt) 2 in the soleus muscle of tail-suspension mice.

These information added in the manuscript (Page 1, 29-32, 37-39; Page 4, Lines 162-167; Page 8, Lines 279-296; Page 10, Lines 358-360, Lines 408-411; Page 11, Lines 408-411).

Figure 6. Effect of dietary AX on mitochondrial biogenesis in SO muscle of tail-suspension mice. The total RNA of SO muscle was extracted and subjected to real-time PCR. Expression ratio relative to that of GAPDH. Data are represented as mean ± S.D. (n=6). Different letters indicate significant differences (P < 0.05) based on ANOVA and Tukey’s test. Real time PCR experiments are representative of at least three independent studies.

In this study, we found that AX showed regulation effect on mitochondrial oxidative phosphorylation and mitochondrial biogenesis in tail-suspension mice. These results prove that the targeted effect of astaxanthin on mitochondria is related to the elimination of mitochondrial ROS, protection of mitochondrial dysfunction, and inhibition of apoptosis caused by mitochondrial dysfunction. However, further studies are necessary to define these mechanisms.

Round 2

Reviewer 1 Report

Good work on this interesting data set and the improvements on the quality of the manuscript.

Author Response

We thank the reviewer for re-evaluating our manuscript. We were pleased to receive the positive evaluation of reviewers, along with their insightful criticisms.

Reviewer 2 Report

The manuscript is ready to be published after proof reading

Author Response

(The authors gave the same response as above.)

Reviewer 3 Report

The authors significantly improved the impact and reliability of the manuscript with the added experiments and added text. 

However, one issue remains given the authors' response. The mere presence of myotubes does not eliminate the presence of myoblasts. Granted I am not as familiar with Sol8 (I work strictly with C2C12), the authors conceded that only 30% of the plate was fused/differentiated myotubes after 96 hours. This leaves a large portion available for the myoblast phenotype which may be skewing your results (unless I am misinterpreting the authors' response). It is recommended that you coat your plates with collagen and differentiate for 5-7 days (7days is preferred). If this is not possible, it is recommended that this be added as a limitation or discussed as a mechanisms that requires the presence of myoblasts.

Author Response

Point-by-point response to the comments of Reviewer

We thank the reviewer for re-evaluating our manuscript. We were pleased to receive the positive evaluation of reviewers, along with their insightful criticisms.

However, one issue remains given the authors' response. The mere presence of myotubes does not eliminate the presence of myoblasts. Granted I am not as familiar with Sol8 (I work strictly with C2C12), the authors conceded that only 30% of the plate was fused/differentiated myotubes after 96 hours. This leaves a large portion available for the myoblast phenotype which may be skewing your results (unless I am misinterpreting the authors' response). It is recommended that you coat your plates with collagen and differentiate for 5-7 days (7days is preferred). If this is not possible, it is recommended that this be added as a limitation or discussed as a mechanisms that requires the presence of myoblasts.

Response:     

We agreed with the comment raised by the reviewer. normally, we have coated the plates with collagen before seeding sol8 cells, but cells were maintained in differentiation medium for 4 days (but not 5-7days) for the formation of myotubes. It was difficult to understand the detailed experimental method for reader. To clarify that experiments with sol8 myotubes are performed under limited conditions containing myoblast, we have explained the method in detail and have described as a mechanism that requires the presence of myoblasts.

These information were added in the manuscript (Page 3, lines 115-120, Page 8-9, lines 300-304, Page 14, lines 531-532). All line numbers mentioned in each response to each comment refer to the numbers that appear on the left margin of the text of the revised manuscript.